# CAUSAL PROCESS MODELS: REFRAMING CAUSAL GRAPH DISCOVERY AS A REINFORCEMENT LEARNING PROBLEM

## ABSTRACT

Most neural models of causality assume static causal graphs, failing to capture the dynamic and sparse nature of physical interactions where causal relationships emerge and dissolve over time. We introduce the Causal Process Framework and its neural implementation, Causal Process Models (CPMs), for learning sparse, time-varying causal graphs from visual observations. Unlike traditional approaches that maintain dense connectivity, our model explicitly constructs causal edges only when objects actively interact, dramatically improving both interpretability and computational efficiency. We achieve this by formulating causal discovery as a multi-agent reinforcement learning problem, where specialized agents sequentially decide which objects are causally connected at each timestep. Our key innovation is a structured representation that factorizes object and force vectors along three learned dimensions (mutability, causal relevance, and control relevance), enabling the automatic discovery of semantically meaningful encodings. We demonstrate that a CPM significantly outperforms dense graph baselines on physical prediction tasks, particularly for longer horizons and varying object counts.

## 1 INTRODUCTION

Causality plays a fundamental role in building intelligent systems capable of physical reasoning (Gerstenberg et al., 2020). Explicitly modeling causal relationships is increasingly recognized to be crucial for developing robust, generalizable, and interpretable neural network models capable of accurate prediction and effective intervention (Xia et al., 2021). Despite their black-box nature, models such as transformers have demonstrated surprising capacity for causal reasoning (Nichani et al., 2024; Shou et al., 2023; Melnychuk et al., 2022; Dettki et al., 2025). One explanation posits that this is possible due to the attention mechanism forming implicit causal edges between tokens (Vaswani et al., 2017; Rohekar et al., 2023). However, recent work has highlighted a phenomenon known as *over-squashing*, in which the attention mechanism (and related message-passing mechanisms in

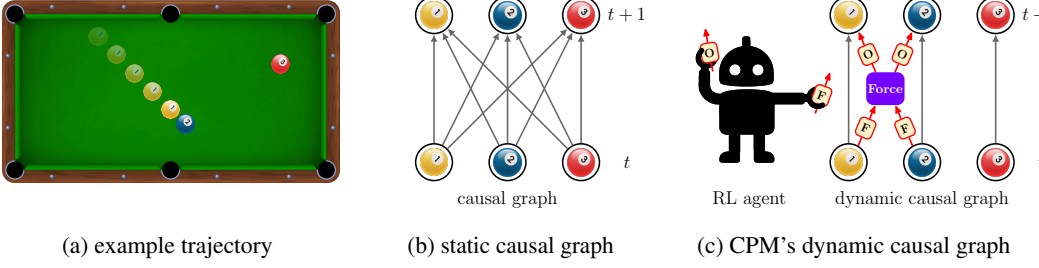

|  |  |  |
|---|---|---|
| (a) example trajectory | (b) static causal graph | (c) CPM's dynamic causal graph |

Figure 1: **Dynamic Causality**: (a) In many physical domains, such as a game of billiards, objects interact only sparsely. (b) Static causal graphs must encode *all possible* interactions, resulting in dense connectivity that fails to capture this local sparsity. (c) In a Causal Process Model (CPM), an RL agent dynamically constructs a causal graph by connecting forces and objects through process blocks, yielding a sparse, dynamic causal graph that reflects the actual interactions at each timestep.

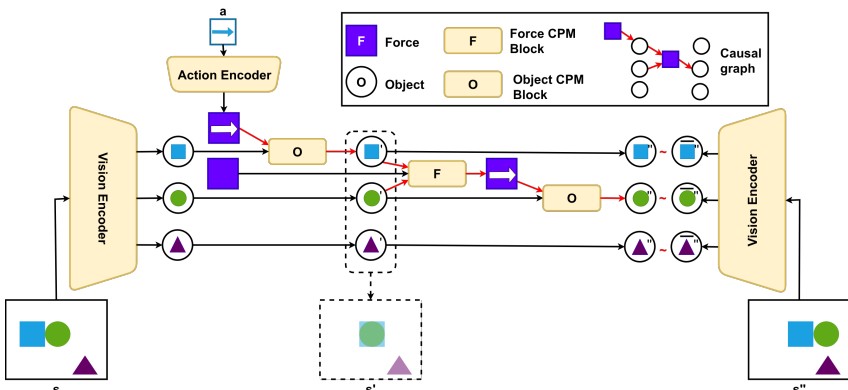

Figure 2: **Model Overview**: Our model has three components: a vision encoder, an action encoder, and a transition function. The transition function is an implementation of a *Causal Process Model*. The state is factorized into distinct object representations, actions are mapped to force representations that act as causal interventions, and the directed edges are causal.

Graph Neural Networks) loses sensitivity to individual tokens or nodes (Barbero et al., 2024a; Alon & Yahav, 2021; Barbero et al., 2024b; Giovanni et al., 2023; 2024; Topping et al., 2022; Scarselli et al., 2009; Battaglia et al., 2018). This compression of information in transformer models can sever causal chains, thus limiting the effectiveness of causal inference.

In contrast, graphical causal models, such as Pearl's Structural Causal Models (SCMs; Pearl, 2009), explicitly encode causal relationships and thus preserve perfect causal connectivity by design. Yet, a key challenge for SCMs is *causal discovery* (Schölkopf et al., 2021): inferring the causal graph from data. Most existing approaches assume access to a complete dataset and construct a static causal graph, e.g., for all possible interactions of three billiard balls a dense graph is necessary (see Fig. 1a). This assumption is misaligned with the nature of physical environments, where causal influence is typically local in space and sparse in time (Butz, 2017; Pitis et al., 2020; Seitzer et al., 2021; Gumbsch et al., 2021; Lange & Kording, 2025). For instance, objects may only interact upon contact. Recent work has therefore emphasized the importance of *local causal models* (Pitis et al., 2020; Seitzer et al., 2021; Urpí et al., 2024; Lei et al., 2024; Willig et al., 2025) that explain causal connections through the sparsest possible graph, which changes dynamically over time.

Our work aims to bridge these areas by proposing a novel causal framework tailored to capture the dynamics of physical object interactions. We propose **Causal Process Models** (CPMs), as a neural implementation of this framework **casting the construction of sparse dynamic causal graphs as a sequential reinforcement learning (RL) problem**. Instead of relying on dense message passing (e.g., soft attention or standard GNNs, Fig. 1b), CPMs use RL agents to dynamically determine all-or-nothing connections between entities (Fig. 1c). This allows the model to adaptively control connectivity based on the input, avoiding the over-squashing problem and enabling more efficient and interpretable causal reasoning.

Our novel causal framework is designed specifically for modeling the dynamics of physical object interactions, aiming to synthesize the formal rigor of *static dependency* theories, e.g. Pearl's do-calculus (Pearl, 2009), with the intuitive strengths of *process-based* accounts (Russell, 1948; Salmon, 1984; Skyrms, 1981; Dowe, 2000, see Section 2 below). Our approach explicitly addresses the limitations of Pearlian SCMs by enabling the construction of sparse, time-varying causal graphs that reflect only the active interactions between objects. When modeling two colliding balls for instance, our framework only instantiates a direct causal link between the balls upon contact, for the transfer of momentum, while leaving them causally disconnected otherwise. This yields a computationally efficient model, only scaling with actual rather than all potential interactions, and one that is highly interpretable since the causal graph mirrors intuitive physical processes.

Our main contributions are: 1) We formalize a *Causal Process Framework* (CPF) for local causal modeling in physical environments. 2) We implement this in a neural architecture as a *Causal Process Model* (CPM) to dynamically infer sparse, time-varying causal graphs by framing edge selection as an RL problem. 3) We apply our CPM to physical interaction scenarios, demonstrating superior performance, interpretability, and scalability compared to densely connected models.

## 2 RELATED WORK

### 2.1 CAUSAL FRAMEWORKS

Pearl's (2009) framework of Structural Causal Models (SCMs) is a dominant approach to causal modeling, by representing causal relationships using directed acyclic graphs (DAGs). An SCM can be described as a tuple $\mathfrak{C} := (\mathbf{S}, \mathbb{P}(\mathbf{U}))$ where $\mathbb{P}$ is a distribution over the exogenous variables $\mathbf{U}$ (i.e., variables external to the system and not caused by any variable within it) and $\mathbf{S}$ is a collection of structural equations of the form:

$$V_i = f_{V_i}(\mathbf{Pa}_{V_i}, \mathbf{U}_{V_i}).$$

Each endogenous variable $V_i$ is determined by a function of its parent variables $\mathbf{Pa}_{V_i}$ (i.e., other variables in the system that directly influence $V_i$) and its associated exogenous noise term $\mathbf{U}_{V_i}$.

While successful in many domains, standard SCMs require extensions to handle systems characterized by dynamic object interactions; without such extensions, they fail to adequately capture the temporal and structural intricacies of such systems (Rubenstein et al., 2016; Weber, 2016; Blom et al., 2020; Boeken & Mooij, 2024). Consider the simple scenario of two colliding balls shown in Fig. 1a. Representing this within a traditional SCM framework often requires specifying potential causal links between all properties of all objects at all relevant timescales. This leads to densely connected causal graphs (Fig. 1b), with the number of causal edges scaling quadratically with time, even when interactions are sparse in reality. Such dense representations suffer from high computational costs for inference and learning, and crucially, obscure the underlying causal structure, hindering interpretability. Thus, a core challenge is to adapt standard SCMs to dynamically represent only the relevant interactions as they occur, rather than needing to specify all potential dependencies.

Recognizing these limitations, other lines of research offer valuable perspectives, often aligning closely with *causal process theories* (Russell, 1948; Salmon, 1984; Skyrms, 1981; Dowe, 2000). Research in cognitive science, such as Gerstenberg et al. (2020)'s counterfactual simulation models, leverage simulation to assess causality and responsibility in physical events, capturing process-like intuitions. Furthermore, philosophical inquiries into causal processes provide rich conceptual foundations, distinguishing *causal* processes from *pseudo*-processes by focusing on mechanisms like causal lines (Russell, 1948), defining causality in ontological terms (Salmon, 1984), or using conserved quantities (Skyrms, 1981; Dowe, 2000). However, this philosophical tradition lacks the computational formalism required for direct implementation in ML systems. Our Causal Process Framework bridges this gap by providing a computationally tractable formalism that integrates process-based intuitions with graphical causal models, enabling dynamic and sparse representations suitable for learning from visual data in physical environments.

### 2.2 NEURAL CAUSAL MODELS

While philosophical causal process theories offer intuitive insights into dynamic physical interactions, their abstract nature limits direct application in scalable machine learning systems. To operationalize these ideas computationally, researchers have sought to integrate causal process intuitions with neural architectures, particularly by embedding SCMs into deep learning frameworks. Previous attempts to reconcile deep learning with SCMs have resulted in Neural Causal Models (NCMs), which model $f_{V_i}$ as feedforward neural nets parametrized by $\theta_{V_i}$ (Xia et al., 2021). Yet this solution still suffers from the disadvantage of needing to train arbitrarily many feedforward neural networks for each node across time. To address this parameter explosion, Zecevic et al. (2021) have tried to theoretically quantify the capacity for GNNs to implement SCMs, but are restricted to the assumption of static causal graph. In contrast, Melnychuk et al. (2022) designed a Causal Transformer that incorporates temporal dynamics to infer causality over time, yet is still unable to yield interpretable graph representations. This limitation arises from its reliance on the potential outcomes framework (Rubin, 1978; Robins & Hernan, 2008), which focuses on estimating counterfactual outcomes without explicitly representing causal relationships as graphs, thus making it less suitable for discovering and utilizing sparse, time-varying structures.

## 2.3 CAUSAL REINFORCEMENT LEARNING

Buesing et al. (2019) have tried to take advantage of the Pearlian causality framework by reformulating the MDP graph as an SCM using which they designed a counterfactually-guided policy search. A similar approach has been pursued by Gasse et al. (2023) in which they draw parallels between confounding variables and offline RL. Neither of these approaches factors the MDP state space into distinct object-centric nodes and their causal relations, instead focusing on the aforementioned inherent causality of the MDP structure as suggested by Bareinboim et al. (2021).

# 3 CAUSAL PROCESS FRAMEWORK

Pearl's structural causal models (SCMs) and do-calculus (Pearl, 2009) provide a powerful foundation for causal reasoning. However, without extensions, it is not straightforward to apply SCM to dynamic physical systems requiring object-centric representations and real-time causal interactions. Prior approaches (Buesing et al., 2019; Gasse et al., 2023) have attempted to bridge model-based RL and causality by representing the full Markov Decision Process (MDP) state $s^t$ using a single node and modeling actions as direct interventions in a static causal graph. However, this approach is limited because it circumvents the problem of inferring the causal structure that generates the underlying environment dynamics (i.e., the causal context; Butz et al., 2025), and focuses only on the causal implications of action sequences.

## 3.1 CAUSAL PROCESS MODELS (CPMS)

To address the inability of SCMs to capture sparse, time-varying interactions, the computational burden of dense connectivity, and the loss of causal information in over-squashed message-passing, we introduce Causal Process Models (CPMs). CPMs dynamically construct sparse causal graphs that represent only active interactions, enabling both computational efficiency and interpretable causal structure in physical environments.

We adopt an *object-centric factorization* of states, in which physical objects are represented separately as nodes $\mathcal{O} = \{O_1, O_2, ..., O_N\}$ (e.g., balls) and interactions between objects are represented as force nodes $\mathcal{F} = \{F_1, F_2, ..., F_M\}$ (e.g., collisions). At each timestep $t$, we have object states $\mathcal{O}^t = \{O_1^t, ..., O_N^t\}$ and force states $\mathcal{F}^t = \{F_1^t, ..., F_M^t\}$. The key insight is that not all objects interact at all times, hence we need to dynamically determine which causal edges are active.

### 3.1.1 DYNAMIC CAUSAL GRAPH CONSTRUCTION

To dynamically determine active causal edges in the graph, CPMs employ two types of specialized controller functions: *interaction scope controllers* $\rho_{\mathcal{O}}^t$ determine which objects interact, that is, exchange forces (e.g., based on spatial proximity), while *effect attribution controllers* $\rho_{\mathcal{O} \leftrightarrow \mathcal{F}}^t$ determine how objects are affected by these interacting forces.

Formally, each controller outputs probabilistic distributions over possible edge subsets at each timestep. The interaction scope controllers $\rho_{\mathcal{O}}^t$ define a distribution over edge sets $J^t \subseteq \mathcal{O}^t \times \mathcal{F}^t$ conditioned on current object states $\mathcal{O}^t$, yielding $J^t \sim \rho_{\mathcal{O}}^t(\cdot \mid \mathcal{O}^t)$. Similarly, the effect attribution controllers $\rho_{\mathcal{O} \leftrightarrow \mathcal{F}}^t$ define a distribution over edge sets $I^t \subseteq \mathcal{F}^t \times \mathcal{O}^{t+1}$ conditioned on current object and force states $(\mathcal{O}^t, \mathcal{F}^t)$, yielding $I^t \sim \rho_{\mathcal{O} \leftrightarrow \mathcal{F}}^t(\cdot \mid \mathcal{O}^t, \mathcal{F}^t)$.

Within this framework, state evolutions are governed by object and force update functions $f_O$ and $f_F$, which propagate information along the dynamically selected causal edges:

$$
\begin{aligned}
\text{Forces} \quad & F_j^t := f_F\left(F_j^{t-1}, \left\{O_i^{t-1}\right\}_{i|(i,j) \in J^{t-1}}\right) \quad & \text{s.t. } J^{t-1} \sim \rho_{\mathcal{O}}^{t-1}, \\
\text{Objects} \quad & O_i^t := f_O\left(O_i^{t-1}, \left\{F_j^t\right\}_{j|(j,i) \in I^{t-1}}\right) \quad & \text{s.t. } I^{t-1} \sim \rho_{\mathcal{O} \leftrightarrow \mathcal{F}}^{t-1}.
\end{aligned}
\tag{1}
$$

Thus, update functions $f_F$ and $f_O$ are force- and object-specific (respectively) and invariant to the number of inputs (i.e., size of the parent node set). When interventions occur at time step $\tilde{t}$, they introduce perturbations over object nodes, denoted as $\text{act}(O_*^{\tilde{t}})$ representing externally applied actions (e.g., hitting a billiard ball). Intuitively, this step captures how such external influences propagate

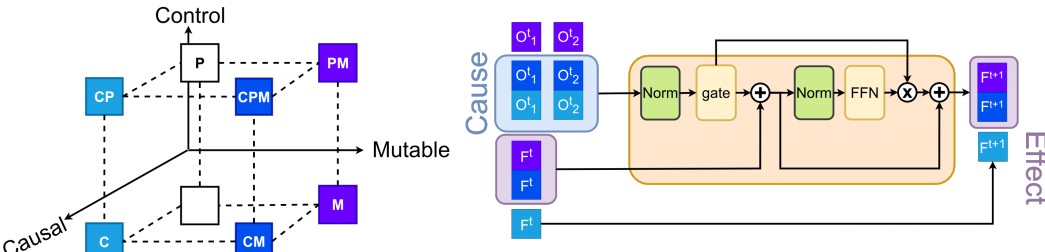

Figure 3: **Causal Process Block** (illustrating $f_F$): Modified transformer with attention replaced by gate mechanism (see Appendix B.3). Incoming causes $O_i^t$ are pre-selected by the causal controllers. Latent vectors are divided into causal (C), control (P), and mutable (M) regions, enforcing structured updates.

forward in time, akin to resimulating the physical system from the intervention point onward: the model dynamically recomputes the subgraph for all subsequent time steps $t \geq \tilde{t}$ by resampling causal edges and updating node states via the controllers and transition functions, ensuring the causal graph reflects the altered dynamics (following Algorithm 1).

### 3.1.2 CONCRETE EXAMPLE: TWO COLLIDING BALLS

To illustrate, consider a scenario with three balls, where Ball 1 collides with Ball 2 at time $t$ (Fig. 1a). The objects are represented as $\{O_1^t, O_2^t, O_3^t\}$, and a single force node $F_1^t$ mediates the interaction. Here, the interaction scope controller $\rho_{\mathcal{O}}^t$ assigns high probability to the edges $J^t = \{E(O_1^t, F_1^t), E(O_2^t, F_1^t)\}$, indicating that both Ball 1 and Ball 2 contribute to generating the collision force. Similarly, the effect attribution controller $\rho_{\mathcal{O} \leftrightarrow \mathcal{F}}^t$ assigns high probability to the edges $I^t = \{E(F_1^t, O_1^{t+1}), E(F_1^t, O_2^{t+1})\}$, specifying that the force affects both balls post-collision. The resulting graph is illustrated in Fig. 1c. In contrast, when the balls are far apart and no interaction occurs, both controllers would output empty edge sets, resulting in a sparse graph with only self-connections during that time period (e.g., Ball 3 in Fig. 1c).

### 3.1.3 INDUCTIVE BIASES

To ground the flexible graph construction in realistic physical principles and mitigate the risk of overfitting to spurious connections, we incorporate two key inductive biases that reflect common patterns in object interactions. **Pairwise Interactions** restrict each force node to connect to exactly two different object nodes. This corresponds to the assumption that typically not more than two objects interact at a certain time step. This restriction can be lifted later to generalize to hypergraphs for more complicated systems (e.g., 3-body problems). **Newton's Third Law and Force Symmetry** are modeled through our mirroring constraint: when two objects interact, the force node must affect both objects that contributed to it. In physical collisions, forces come in equal and opposite pairs acting on both objects. This design ensures physical consistency while maintaining computational efficiency. In environments with asymmetric interactions (e.g., large objects unaffected by small ones), the learned weights in $f_O$ can effectively set the influence to zero (see Appendix B.1 for formal definitions).

## 4 MODEL

We base our model implementation on the Contrastively-trained Structured World Model (C-SWM; Kipf et al., 2020). The model consists of an *object-centric vision encoder*, an *action encoder*, and a *transition function* (Fig. 2). We keep the structure of the vision and action encoders intact, but modify the transition function.

The *vision encoder* is a CNN-based object extractor $E_{\text{ext}}$, operating directly on images and outputting $I$ feature maps. Each feature map $m_i^t = [E_{\text{ext}}(s^t)]_i$ acts as an object mask where $[\dots]_i$ is the selection of the $i^{\text{th}}$ feature map. An MLP-based object encoder $E_{\text{enc}}$ with shared weights across objects maps the flattened feature map $m_i^t$ to object latent representation: $O_i^t = E_{\text{enc}}(m_i^t)$. Additionally, an MLP-based *action encoder* maps action $a^t$ to force latent representation: $F^t = A(a^t)$. Next, we

introduce our new transition function (Sec. 4.1) before detailing how to construct the causal graph on the fly using reinforcement learning (Sec. 4.2).

## 4.1 CAUSAL PROCESS BLOCK

Our main innovation is the *Causal Process Block* as a neural network implementation of a CPM (Fig. 3). Before introducing the technical details, we need to address a key challenge: not all components of force and object representations play the same role in causal interactions.

### 4.1.1 STRUCTURED REPRESENTATIONS

Let us revisit the example of collision between two balls (Fig. 1a). Their masses affect momentum transfer (causally relevant) but remain unchanged during the collision (immutable). In contrast, their velocities are both causally relevant and mutable. Meanwhile, visual properties like color may change due to lighting, but don't affect the collision dynamics (mutable but not causally relevant). To capture these distinctions and enable our model to learn interpretable encodings that naturally separate these different physical properties, we factorize our representations along three key binary subspaces of *Causal Relevance* ($C$), *Control Relevance* ($P$), and *Mutability* ($M$). The binary nature of each subspace arises from selective routing within the CPM (Fig. 3).

Causal Relevance ($C$) describes whether a component influences the dynamics of other objects. For instance mass and velocity affect collision outcomes ($C = 1$), but color does not ($C = 0$). Control Relevance ($P$) encodes whether a component is used by the control/policy functions for decision-making. For example, a controller deciding which balls are about to collide will rely on current positions and velocities ($P = 1$), but will ignore other properties such as mass or purely visual features like color ($P = 0$). Mutability ($M$) captures whether a component can change over time through interactions. For instance, an object being struck may change velocity ($M = 1$), while its mass remains constant ($M = 0$). Importantly, while the factorization structure is fixed (based on $C$, $P$, and $M$), the model *learns* which specific features belong to each category through training. This hard architectural partitioning forces the model to discover semantically meaningful, disentangled representations that align with intuitive physical concepts.

### 4.1.2 TECHNICAL IMPLEMENTATION

We use two feedforward neural networks, $f_F(\ldots; \theta_F)$ and $f_O(\ldots; \theta_O)$, shared by all the force and object nodes respectively. The force vector $F_j^t := \bigoplus_{C,P,M \in \{1,0\}, (C,P,M) \neq (0,0,0)} F_j^{t,CPM}$ is the concatenation of all combinations of the $C$, $P$, $M$ dimensions except for $(C,P,M) = (0,0,0)$, which must be omitted since forces, by definition, do not contain subspaces that are irrelevant to the causal process. This results in $2^3 - 1 = 7$ sub-vectors of equal size $d_F$, where a sub-vector's identity determines how it is processed by the neural networks: The object-update function $f_O$ and force-update function $f_F$ operate exclusively on the causally relevant subspace ($C = 1$), while mutable parts are updated and immutable parts are copied unchanged ($M = 0$; Fig. 3). The object vector $O_i^t$ is more straightforwardly divided into $2^3 = 8$ subvectors, i.e., all possible combinations of the $C$, $P$, $M$ dimensions, including the $(C,P,M) = (0,0,0)$ subspace: $O_i^t := \bigoplus_{C,P,M \in \{1,0\}} O_i^{t,CPM}$. The extra subspace is present here for the network to learn to shift visual input features that are irrelevant to the causal process into this subspace, for example the object's color in a collision event (See Appendices B.2 and B.3 for details). Note that the implementation of the causal process blocks $f_O(\cdots | \theta_O)$ and $f_F(\cdots | \theta_F)$ is similar to that of transformer blocks, but with the attention mechanism (Vaswani et al., 2017) replaced by indices of the chosen force and object nodes (tokens in transformers; see Appendix B.3 and Fig. 3 for more details). Unlike the attention mechanism of the transformer, $I^t, J^t$) can also be an empty set. This is analogous to transformer attention assigning zero weight to all the tokens, which they cannot do by design.

## 4.2 CAUSAL CONTROLLER

The main proposal of our model is that we perform graph construction through sequential decision making using the interaction scope and effect attribution controllers. We treat causal discovery as a multi-agent RL problem. One agent (the interaction scope policy $\pi_O(\mathcal{O}^t) := \rho_\mathcal{O}^t$) determines the

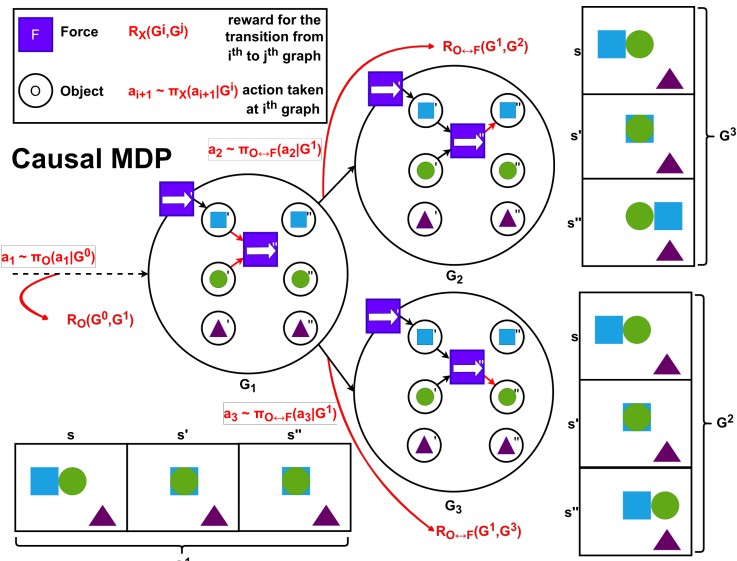

Figure 4: **Causal MDP** used by the reactive agents to construct causal process graphs. Agents successively add edges to the causal graph. Each causal graph hypothesis corresponds to a potential sequence of frames.

scope of interacting objects, and another (the effect attribution policy $\pi_{O\leftrightarrow F}\left(\mathcal{O}^t, \mathcal{F}^{t+1}\right) := \rho^t_{\mathcal{O}\leftrightarrow\mathcal{F}}$) determines how force effects are attributed .

More specifically, the chosen indices $I^t$ and $J^t$ are provided by the agents $\pi_{\mathcal{O}\leftrightarrow\mathcal{F}}$ and $\pi_{\mathcal{O}}$. The two agents alternate outputting an action. An action taken by $\pi_{\mathcal{O}}$ corresponds to two edge additions $E\left(O^t_i, F^{t+1}\right), E\left(O^t_j, F^{t+1}\right), i \neq j$ to the graph (selecting a pair of objects for interaction). Whereas an action taken by $\pi_{\mathcal{O}\leftrightarrow\mathcal{F}}$ results in either one or two edge additions $E\left(F^t, O^t_i\right), E\left(F^t, O^t_j\right)$ (attributing the force effect to either one or both objects; see Fig. 4). The index set $I^t$ is sampled using the policy of the agent $\pi_{\mathcal{O}\leftrightarrow\mathcal{F}}$. Note that the policies only utilize the Control Relevant (P) features of the latent representations:

$$
\begin{aligned}
I^t \sim \pi_{\mathcal{O}\leftrightarrow\mathcal{F}}\left(I^t \mid G^t, W_O, W_F\right) &= \mathrm{softmax}\left(Q_{\mathcal{O}\leftrightarrow\mathcal{F}}\left(G^t, I^t \mid W_O, W_F\right)\right) \\
&= \mathrm{softmax}\left(\frac{\left(F^{t,C1M}W_F\right)\left(\left[O^{t,C1M}_i; O^{t,C1M}_j; \left(O^{t,C1M}_i + O^{t,C1M}_j\right)/2\right]W_O\right)^T}{d}\right),
\end{aligned}
\tag{2}
$$

where $W_O \in \mathbb{R}^{4d_O \times d}$, $W_F \in \mathbb{R}^{4d_F \times d}$, and $Q_{\mathcal{O}\leftrightarrow\mathcal{F}}$ is the corresponding Q-value. $J^t$, on the other hand, is sampled using the policy of the agent $\pi_{\mathcal{O}}$:

$$
J^t \sim \pi_{\mathcal{O}}\left(J^t \mid G^t, W_{\tilde{O}}\right) = \sigma\left(Q_{\mathcal{O}}\left(G^t, J^t \mid G^t, W_{\tilde{O}}\right)\right) = \sigma\left(\left(O^{t,C1M}_i W_{\tilde{O}}\right) \cdot \left(O^{t,C1M}_j W_{\tilde{O}}\right)\right),
\tag{3}
$$

where $W_{\tilde{O}} \in \mathbb{R}^{4d_O \times d}$, $\sigma$ is the sigmoid function, and $Q_{\mathcal{O}}$ is the corresponding Q-value.

We then define separate reward functions for $\pi_{\mathcal{O}\leftrightarrow\mathcal{F}}$ and $\pi_{\mathcal{O}}$, modeled by MLPs parameterized by $\theta_{R_{\mathcal{O}\leftrightarrow\mathcal{F}}}$ and $\theta_{R_{\mathcal{O}}}$ respectively:

$$
\begin{aligned}
R_{\mathcal{O}\leftrightarrow\mathcal{F}}\left(G^t, G^{t+1} \mid \theta_{R_{\mathcal{O}\leftrightarrow\mathcal{F}}}\right) &= \mathrm{MLP}\left(G^t_V, \mathbb{1}_{E\left(F^t, O^t_i\right)}, \mathbb{1}_{E\left(F^t, O^t_j\right)}, G^{t+1}_V \Big| \theta_{R_{\mathcal{O}\leftrightarrow\mathcal{F}}}\right), \\
R_{\mathcal{O}}\left(G^t, G^{t+1} \mid \theta_{R_{\mathcal{O}}}\right) &= \mathrm{MLP}\left(G^t_V, \mathbb{1}_{E\left(O^t_i, F^{t+1}\right)\wedge E\left(O^t_j, F^{t+1}\right)}, G^{t+1}_V \Big| \theta_{R_{\mathcal{O}}}\right),
\end{aligned}
\tag{4}
$$

where $G^t := (G^t_V, G^t_E)$ is the graph at time $t$ and $\mathbb{1}_{E(\cdot,\cdot)}$ indicates the presence or absence of the edge $E(\cdot,\cdot)$. These reward functions are learned through inverse reinforcement learning (IRL), where the goal is to find a reward function whose corresponding optimal policy would select causal edges that would minimize prediction error.

## 5 TRAINING

### 5.1 TRAINING OVERVIEW AND REWARD LEARNING

Our training procedure addresses a fundamental challenge: jointly learning the causal dynamics model and the policy for selecting causal edges. The key insight is that good edge selections lead to better prediction accuracy, which we use as an implicit reward signal through inverse reinforcement learning (Ng & Russell, 2000).

The optimization has three objectives: 1) The CPM should accurately predict future states given the selected causal edges; 2) the RL agents should select edges that minimize prediction error; 3) the reward functions $R_{\mathcal{O}}$ and $R_{\mathcal{O} \leftrightarrow \mathcal{F}}$ should capture which edge selections lead to better predictions. At convergence, the agents select sparse causal graphs that capture only the active interactions, leading to both computational efficiency and interpretability. The optimum of the combined CPM and RL objectives is reached when the agents consistently construct causal graphs that minimize the prediction error; at this point the learned reward networks stabilize.

### 5.2 TRAINING PROCEDURE

The overall goal is to learn a predictive world model (CPM) whose structure is determined by the policies $(\pi_{\mathcal{O}}, \pi_{\mathcal{O} \leftrightarrow \mathcal{F}})$. This requires optimizing both the model parameters $\Theta$ and the policy parameters $\Psi := [W_O, W_F, W_{\tilde{O}}]$. We achieve this using a 3-stage procedure that also involves expectation-maximization (EM) with alternating optimization (Dempster et al., 1977).

**1. Prediction.** At this stage, we freeze all the weights except $\Theta = [\theta_V; \theta_A; \theta_O; \theta_F]$ and sample edges $\{I^{\tau}, J^{\tau}\}_{\tau}$ using frozen controllers. This allows the vision encoder, action encoder, and transition functions ($f_O, f_F$) to learn useful representations before the controllers get the chance to optimize their behavior on these representations. We train the model parameters $\Theta$ using contrastive loss (Kipf et al., 2020):

$$
\begin{aligned}
\mathcal{L}_{\text{pred}}\left(\Theta | \beta, \mathcal{D}_{\text{pred}}\right) = &\left\| \text{CPM}\left(\text{V}\left(s^t \middle| \theta_V\right), \text{A}\left(a^t \middle| \theta_A\right), \left(I^{\tau}, J^{\tau}\right)_{t_{\tau}} \middle| \theta_O, \theta_F\right) - \text{V}\left(s^{t+1} \middle| \theta_V\right) \right\| \\
&+ \max\left(0, \beta - \left\| \text{V}\left(\tilde{s}^t \middle| \theta_V\right) - \text{V}\left(s^{t+1} \middle| \theta_V\right) \right\|\right)
\end{aligned}
\tag{5}
$$

where $\mathcal{D}_{\text{pred}} = \left\{ \left(s^t, a^t, s^{t+1}\right), \tilde{s}^t, \left\{ \left(G^{t_{\tau}}, I^{t_{\tau}}, J^{t_{\tau}}, G^{t_{\tau}+1}\right)\right\}_{\tau} \right\}_t$, V and A are the vision and action encoders, $\tilde{s}^t$ is a negative example sampled from the experience buffer, and the hinge margin $\beta$ is set to 1 (following Kipf et al., 2020).

**2. Expectation.** In the second stage, we freeze all the weights but $\Theta_R = [\theta_{R_{\mathcal{O} \leftrightarrow \mathcal{F}}}; \theta_{R_{\mathcal{O}}}]$ and use temporal difference (TD) loss (Watkins & Dayan, 1992) to learn reward functions:

$$
\mathcal{L}_{\text{TD}}\left(\Theta_R \mid \mathcal{D}_{\text{TD}}, \Psi\right) = \sum_{X \in \{\mathcal{O} \leftrightarrow \mathcal{F}, \mathcal{O}\}} \sum_{\tau} \left\| \underbrace{R_X\left(G^{\tau}, G^{\tau+1} \middle| \theta_{R_X}\right)}_{\text{learned}} - \underbrace{\left(Q_X\left(G^{\tau}, I_X^{\tau} \mid \Psi\right) - \gamma \max_{I_X^{\tau+1}} Q_X\left(G^{\tau+1}, I_X^{\tau+1} \mid \Psi\right)\right)}_{\text{target}} \right\|
\tag{6}
$$

where $\mathcal{D}_{\text{TD}} = \left\{ \left(G^{\tau}, I_{\mathcal{O}}^{\tau}, I_{\mathcal{O} \leftrightarrow \mathcal{F}}^{\tau}, G^{\tau+1}\right)\right\}_{\tau}$ and we set $\gamma = 0.9$.

**3. Maximization.** During the third stage, we freeze all but the policy parameters $\Psi$ and use the same TD loss from above with target and learned terms reversed. The agents learn to select edges based on the rewards $R_{\mathcal{O} \leftrightarrow \mathcal{F}}$ and $R_{\mathcal{O}}$.

The overall optimization landscape is complex; the optimum represents a state where the agents select the sparse causal graph that yields the minimal prediction loss for the CPM, and simultaneously, the reward MLPs stabilize under the IRL objective.

## 6 EXPERIMENTS

We hypothesize that our model outperforms models that assume dense causal graphs to capture physical interactions in: 1) longer prediction horizons; 2) test-time generalization across unobservable

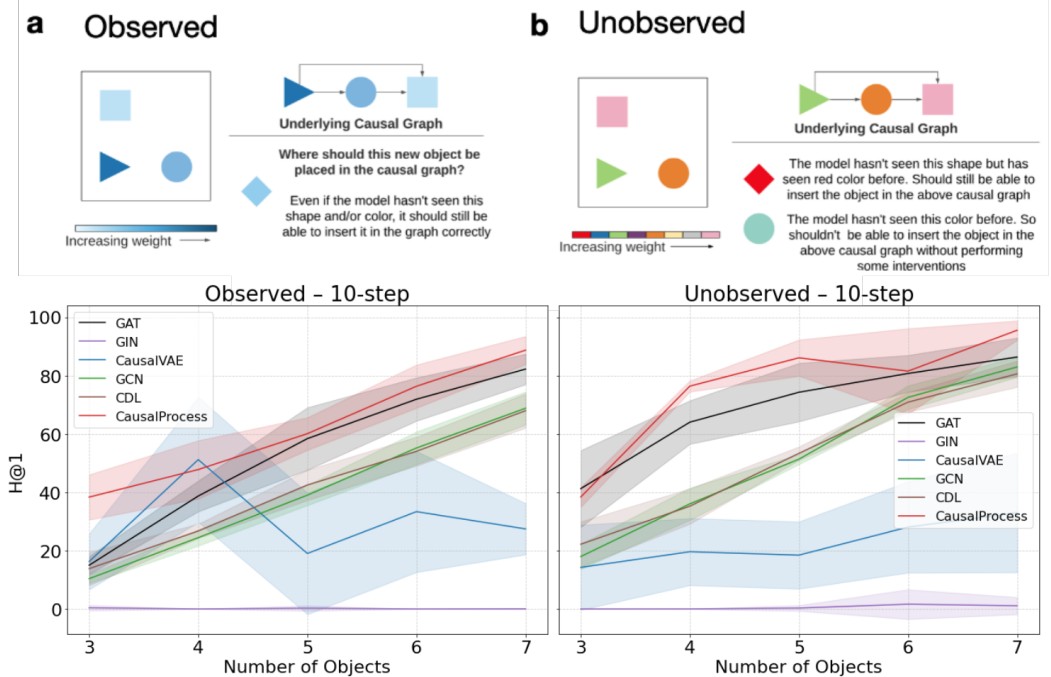

Figure 5: **Prediction results for a synthetic physics environment** in a) observed and b) unobserved settings (Ke et al., 2021). **Top:** Description of the task. **Bottom:** Prediction metric vs number of objects after 10 steps (average of 10 seeds).

properties; 3) robustness with regards to the number of objects in the scene; 4) solving downstream tasks. We use the *physics environment* designed by Ke et al. (2021) to empirically answer these questions (Fig. 5 top). The environment consists of different objects colored according to their weights. The only force in this environment is pushing (double-pushes are not allowed) and only heavier objects can push lighter ones. The environment has two settings: an *observed* setting (Fig. 5a) where weight corresponds to the intensity of a particular color and an *unobserved* setting (Fig. 5b) where different colors did not systematically map to different weights.

## 6.1 COMPARISON BASELINES

We compare our model against 10 baselines, a graph attention network (GAT) (Veličković et al., 2018), a graph isomorphism network (GIN) (Xu et al., 2019), a causal variatiational auto-encoder (CausalVAE) (Yang et al., 2021), a graph convolutional network (GCN) (Kipf & Welling, 2017), a causal dynamics learning network (CDL) (Wang et al., 2022), a graph neural network (GNN) (Scarselli et al., 2009), a transformer network (Vaswani et al., 2017), a recurrent independent mechanisms (RIM) network (Goyal et al., 2021b), a schema / object-file factorization network (SCOFF) (Goyal et al., 2021a), and a modular network which has a separate MLP to model each object's dynamics (Ke et al., 2021).

## 6.2 PREDICTION METRICS

To investigate robustness towards the length of prediction horizons, we trained the model to make 1-step predictions in the *Observed* setting with 5 objects and then tested for 5 (Fig. 9) and 10 steps (Fig. 5a bottom, Fig. 7). We used Hits at Rank 1 (H@1) to measure model performance as an all-or-nothing metric measuring how often the rank of the predicted representation was 1 when ranked against all reference state representations. Here, our model broadly outperformed the baseline models, with the gap increasing over longer time horizons.

Next, to estimate the test-time generalization across unobservable properties, we trained our model in the *Unobserved* setting where generalization at test time is harder due to previously unseen weights.

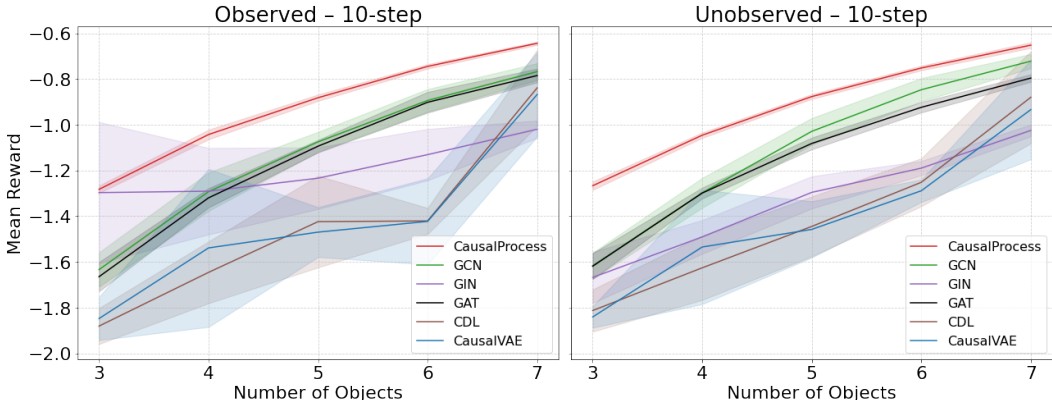

Figure 6: **Downstream RL results** over number of objects. Mean reward vs number of objects. All results are the average of 10 seeds

Again, our model broadly outperformed the baselines displaying capacity to generalize also in this domain (Fig. 5b bottom; see Fig. 7 and Fig. 9 for more results).

### 6.3 DOWNSTREAM RL TASKS

To make sure the above metrics overlap with the learned model's usefulness for downstream tasks, we also tested our CPM's capacity to serve as a world model for a model-based RL agent. The agent's task was to move an object to a certain location each taken step resulting in negative reward. In both Observed and Unobserved settings, the agent with CPM as model of the environment broadly outperformed the baselines for all objects in 10-step unrolling of the learned model (Fig. 6, Fig. 8).

## 7 DISCUSSION

In this paper, we introduced the Causal Process Framework (CPF) as a novel approach for modeling the dynamics of physical object interactions. Our key contribution is the Causal Process Model (CPM), which implements this framework by treating the edge distributions inherent to CPF as a reinforcement learning policy. Instead of the soft, dense connections typical of many baselines (Veličković et al., 2018; Goyal et al., 2021a; Xu et al., 2019; Yang et al., 2021; Kipf & Welling, 2017; Wang et al., 2022; Scarselli et al., 2009; Vaswani et al., 2017; Goyal et al., 2021b), our model employs RL agents to dynamically construct sparse, time-varying causal graphs. Our experiments in a simulated physics environment (Ke et al., 2021) show that this approach not only improves prediction accuracy and downstream task performance compared to baselines, but also excels in generalization and scalability.

The superior performance of our model, particularly over longer prediction horizons and with a varying number of objects, lends strong support to our central hypothesis. We argue that by explicitly modeling only active causal links, the CPM avoids the pitfalls of dense message-passing architectures (Barbero et al., 2024a; Alon & Yahav, 2021; Barbero et al., 2024b; Giovanni et al., 2023; 2024; Topping et al., 2022; Scarselli et al., 2009; Battaglia et al., 2018). Our discrete, "all-or-nothing" connections, determined by a goal-oriented RL agent, preserve the salience of individual interactions. This leads to more robust and precise world models, which proved crucial for the model-based RL agent's success in downstream tasks. Furthermore, the model's ability to generalize to unobserved object properties suggests that it learns an underlying model of physical dynamics rather than memorizing superficial correlations.

Despite these promising results, the present work has several limitations that open clear avenues for future research. A crucial next step is to deepen the analysis of the learned representations. To do so, the semantic content of the force and object sub-vectors could be decoded to verify that our inductive biases are indeed effective in fostering an interpretable internal structure. To provide further validation our claims of causal discovery, we will be necessary to compare the inferred graphs against the ground-truth interaction graphs of the simulation, providing a quantitative measure of the model's ability to recover the true causal processes.

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

# A  OBJECT-CENTRIC CAUSAL DYNAMICS

Consider two objects $O_1$ and $O_2$ (depicted in magenta) with a force $F$ (depicted in violet) acting on them:

$$O_1 \xrightarrow{F} O_2$$

This recovers the familiar structure of a directed acyclic graphs (DAGs) from Pearl's causal formalism Pearl (2009). However, in physical interactions, such as in a collision, it is not always clear which object is the "cause" since both are affected simultaneously. A more intuitive representation would be a bidirectional edge:

$$O_1 \xleftrightarrow{F} O_2$$

However, DAGs prohibit cycles and bidirectional edges. To resolve this, we introduce *temporal dynamics* which represent causal effects as unfolding over time rather than as a simultaneous influence. Thus, a collision between object $O_1$ and object $O_2$ yields forces $F_2$ and $F_3$ as emerging from the past state and influencing future object states:

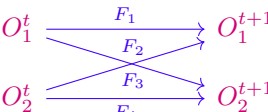

Yet, this representation still has drawbacks. Specifically, we break the identity of the force $F$ into $F_2$ and $F_3$ which, in principle, can act as separate causal links ($F_1$ and $F_4$ can be thought of as inertia). This becomes apparent when interventions are applied. Let us imagine that somebody picks up object $O_1$ just before it collides with $O_2$. This can be represented by a do-calculus-like intervention applied to either $O_1^t$ or $O_1^{t+1}$:

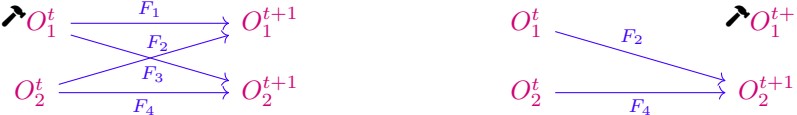

When the intervention is applied to $O_1^t$, the graph structure is preserved, thus implying a no-collision scenario (one of the balls was lifted). Yet, the same graph can also imply a collision scenario. This kind of setup necessitates having causal links between objects that can potentially collide irrespective of the actualization of said collision. While this approach can work in principle, it results in extremely dense graphs with complete subgraphs per time step, especially in cluttered scenes. Ideally, we would like to have causal links in our graph if there is an actualized interaction between the involved objects.

On the other hand, intervening on $O_1^{t+1}$ results in a graph a with counter-intuitive interpretation: $O_1$ gets lifted at time step $t + 1$, while $O_2$ behaves as if a collision has happened. This is due to the split of $F$ into $F_2$ and $F_3$ since an intervention removes $F_3$ while leaving $F_2$ untouched. To tackle the aforementioned issues, let us re-imagine force edges as nodes and re-introduce $F_2$ and $F_3$ as a single node $F$ and extend the time horizon by a step.

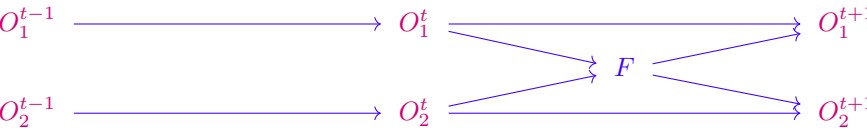

Now, imagine, just like before, the ball $O_1$ gets picked up at time step $t$. In do-calculus terms, this amounts to intervention to $O_1^t$ which results in mutilation of the edge $O_1^{t-1} \rightarrow O_1^t$

While the problem of splitting of the force identity seems to be resolved here, the graph structure modeling the collision remains preserved despite the intervention. As mentioned before, this can be addressed by complete subgraphs per time step, which is not desirable for our purposes. This problem arises due to the inclusion of time dynamics into our graphs. Unlike in Pearlian Causality, in physics, interventions at a time step have implications for the causal connections corresponding to downstream time steps. To account for that, we have to re-imagine interventions under a new framework that takes physical processes and time into account (see Algorithm 1).

## B  MODEL DETAILS

### B.1  INDUCTIVE BIAS

We introduce two inductive biases: (1) limiting each force node to interact with exactly two objects to reflect pairwise interactions, and (2) enforcing a bidirectional mirroring constraint to ensure temporal coherence in causal attribution. Formally the latter is defined as:

$$\forall i, j, k, t : \left\{ E(O_i^t, F_j^{t+1}), E(O_k^t, F_j^{t+1}) \right\} \subset G_E^t \implies$$
$$E(F_j^{t+1}, O_i^{t+1}) \in G_E^t \veebar E(F_j^{t+1}, O_k^{t+1}) \in G_E^t,$$
$$\forall i, j, t : E(F_j^{t+1}, O_i^{t+1}) \in G_E^t \implies E(O_i^t, F_j^{t+1}) \in G_E^t.$$

### B.2  VECTOR CONSTRAINTS

*Causal* relevance is coded by $C$. Perturbing the sub-vectors of the parent with $C = 0$ does not affect the child nodes, i.e., only the causally-relevant $C = 1$ sub-vectors affect the child nodes:

$$\forall t, i : F_j^{t+1,1PM} = \widetilde{F}_j^{t+1,1PM} \implies f_O \left( O_i^t, \{F_j^{t+1}\}_{j|(j,i)\in I^t}; \theta_O \right) = f_O \left( O_i^t, \left\{ \widetilde{F}_j^{t+1} \right\}_{j|(j,i)\in I^t}; \theta_O \right).$$

*Control* relevance is coded by $P$. Two force vectors whose sub-vectors with $P = 1$ are identical have identical control functions that are conditioned on them, i.e., control functions are conditioned only on the control-relevant sub-vectors :

$$\forall t : F_j^{t+1,C1M} = \widetilde{F}_j^{t+1,C1M} \implies \rho_{\mathcal{O}\leftrightarrow\widetilde{\mathcal{F}}}^t = \rho_{\mathcal{O}\leftrightarrow\mathcal{F}}^t.$$

Lastly, *mutability* is coded by $M$. If $M = 0$, the corresponding sub-vector does not change over time, i.e., an immutable sub-vector does not change over time: $\forall t, j : F_j^{t,CP0} = F_j^{t+1,CP0}$.

### B.3  CAUSAL PROCESS BLOCK

Given the data $F^t, O_1^t, \ldots, O_n^t$, and the chosen indices $J^t$, we calculate $F^{t+1}$ in the following way:

$$F^{t+1} := f_F \left( F^t, O_1^t, \ldots, O_n^t, J^t \mid \theta_F \right),$$

with $O^{t+1}$ also calculated similarly.

$$\text{gate} := \frac{1}{|J^t|} \sum_{i \in J^t} \left( O_i^{t,1PM} W_{\text{gate}}^F \right) W_{\text{output}}^F,$$

$$\text{residual} := \text{gate} + F^{t,CP1},$$

$$F^{t+1,CP1} := \chi_{\text{gate}\neq 0} \odot \text{FFN} \left( \text{Norm} \left( \text{residual} \right) \right) + \text{residual},$$

$$F^{t+1} := \left[ F^{t+1,CP1}; F^{t,CP0} \right],$$

where $\theta_F := \left\{ W_{\text{gate}}^F, W_{\text{output}}^F, W_1^F, W_2^F, b_1^F, b_2^F \right\}$ FFN is a feed-forward neural network $\text{FFN}(x) := \max\left( 0, x W_1^F + b_1^F \right) W_2^F + b_2^F$, $W_{\text{gate}}^F$ is the analogue of the attention mechanisms value token projection, and $W_{\text{output}}^{\mathcal{F}}$ is again the analogous out-projection that maps the token from the attention dimension back to residual dimension (Vaswani et al., 2017)

## C PLOTS

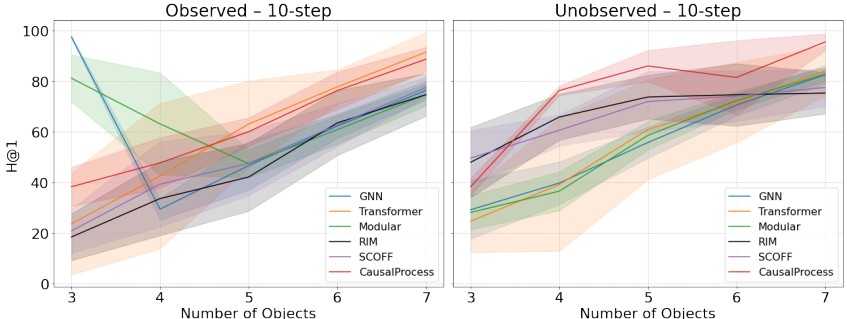

Figure 7: Prediction metric vs number of objects for 10-steps.

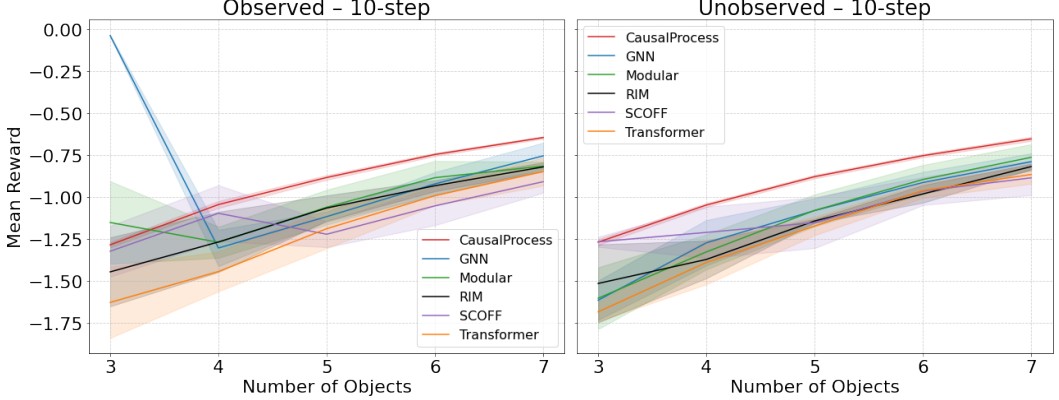

Figure 8: *Downstream RL results over number of objec.* Mean reward vs number of objects. All results are the average of 10 seeds

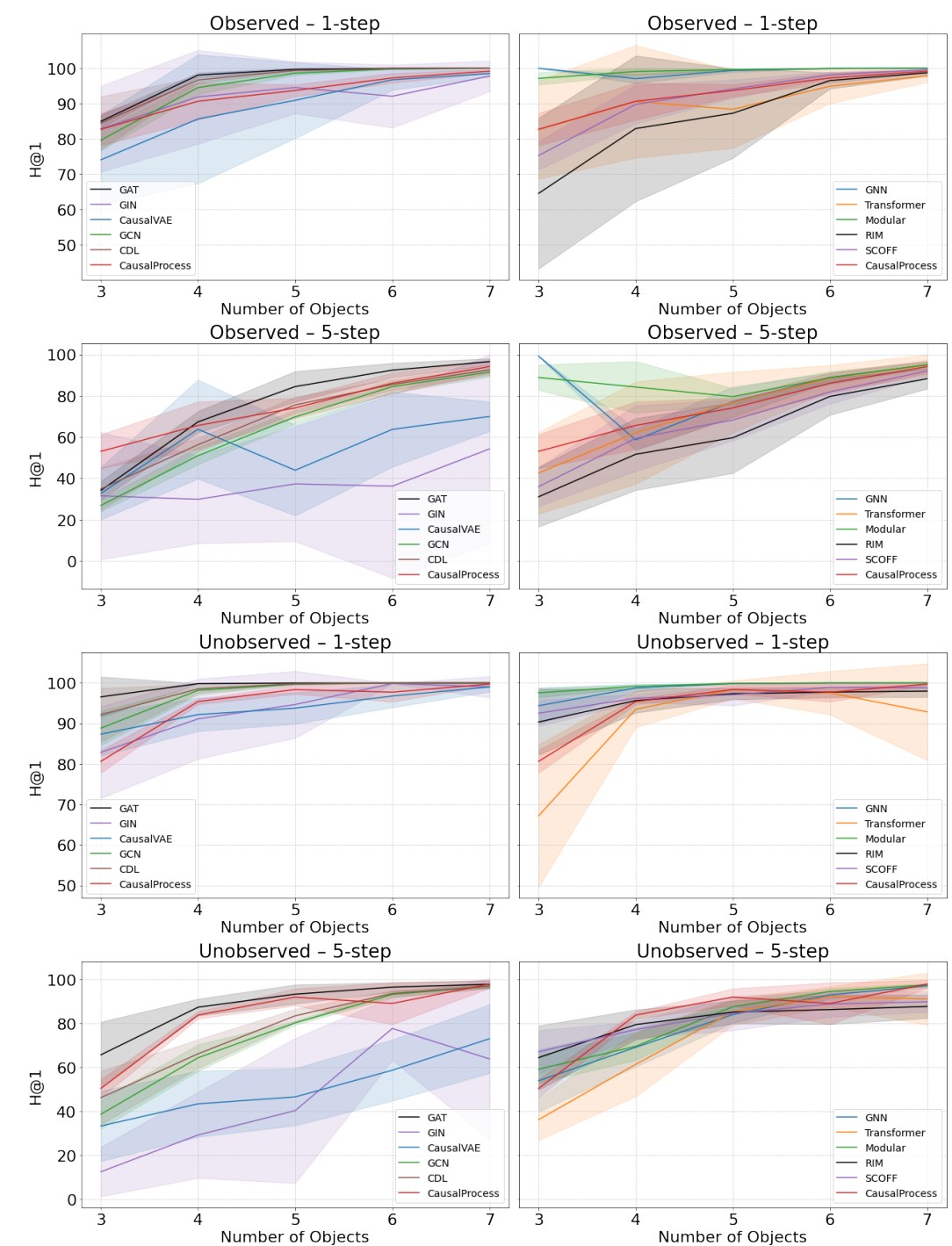

Figure 9: Prediction metric vs number of objects after 1 and 5 steps (average of 10 seeds).

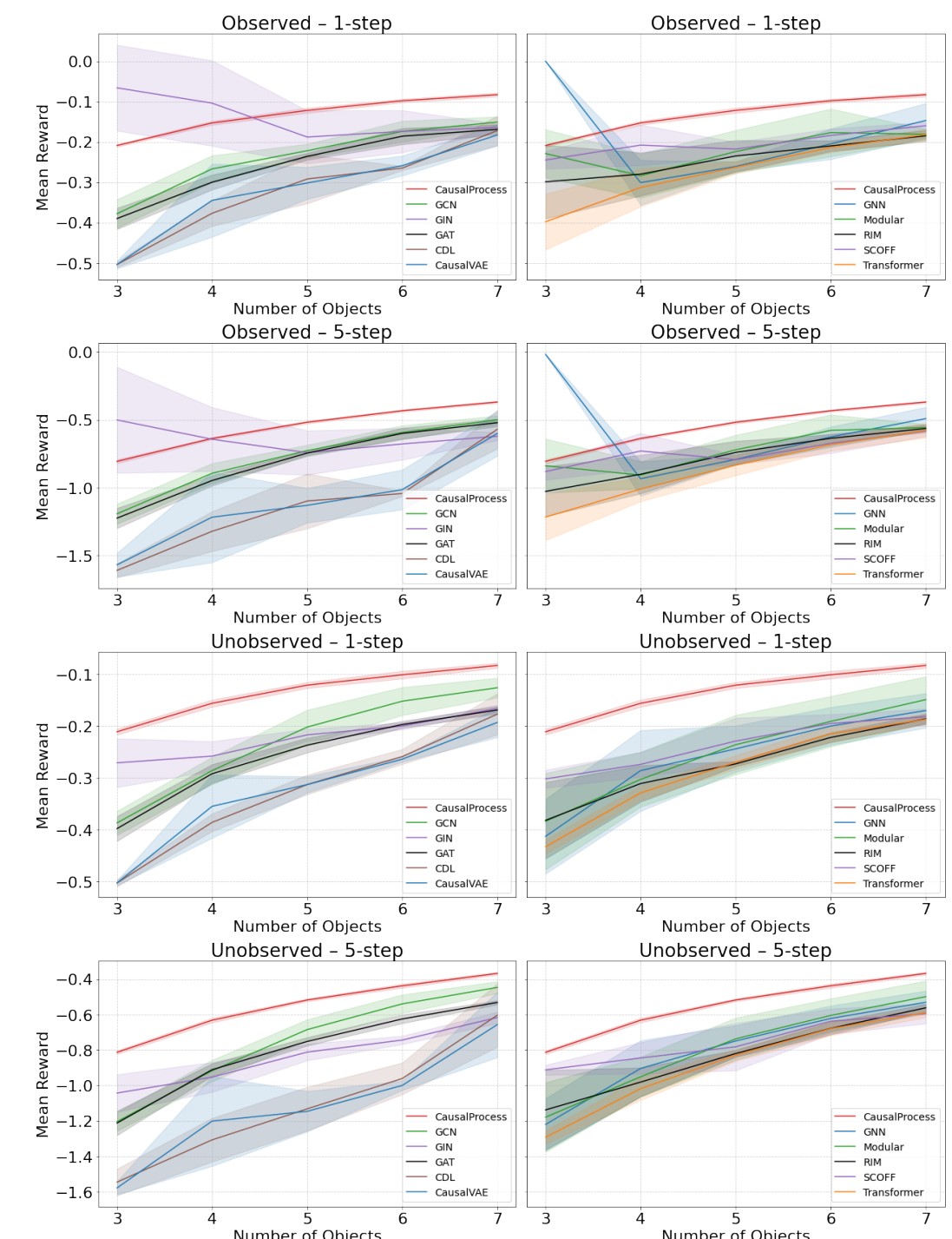

Figure 10: Mean reward vs number of objects after 1 and 5 steps (average of 10 seeds).

# D    ALGORITHMS

---

**Algorithm 1** Interventions under Causal Process Framework

---

**Require:** $\mathcal{F}^t, \mathcal{O}^t, f_F, f_O, \rho^t_{\mathcal{O}\leftrightarrow\mathcal{F}}, \rho^t_{\mathcal{O}}, act\left(O^{\tilde{t}}_*\right), G_{\mathcal{O}^1:\mathcal{O}^T}, t \in \{1,\ldots,T\}, i \in \{1,\ldots,I\}, j \in \{1,\ldots,J\}$

**Ensure:** Output result $\widetilde{G}_{\mathcal{O}^1:\mathcal{O}^T}$

1: Initialize $\widetilde{\mathcal{O}}^{\tilde{t}-1} := \left\{O^{\tilde{t}-1}_*\right\}\dot{\cup}\mathcal{O}^{\tilde{t}-1}$

2: Initialize $\widetilde{\mathcal{F}}^{\tilde{t}} := \mathcal{F}^{\tilde{t}}$

3: Initialize $J^t := \{(i,j)\}$ s.t. $E\left(O^{\tilde{t}-1}_i, F^{\tilde{t}}_j\right) \in G^{\mathcal{E}}_{\mathcal{O}^{\tilde{t}-1}:\mathcal{F}^{\tilde{t}}}$

4: Initialize $\widetilde{G}_{\mathcal{O}^1:\mathcal{O}^{\tilde{t}-1}} := G_{\mathcal{O}^1:\mathcal{O}^{\tilde{t}-1}}$

5: **for** $t = \tilde{t},\ldots,T$ **do**                             ▷ Loop from $t = \tilde{t}$ up to $T$

6:      $\widetilde{G}_{\mathcal{O}^1:\mathcal{F}^t} := \left(\widetilde{\mathcal{F}}^t\dot{\cup}\widetilde{G}^{\mathcal{V}}_{\mathcal{O}^1:\mathcal{O}^{t-1}}, \left\{E\left(\widetilde{O}^{t-1}_i, \widetilde{F}^t_j\right)\right\}_{(i,j)\in J^t}\dot{\cup}\widetilde{G}^{\mathcal{E}}_{\mathcal{O}^1:\mathcal{O}^{t-1}}\right)$    ▷ Update the graph

7:      $I^t \sim \rho^t_{\widetilde{\mathcal{O}}\leftrightarrow\widetilde{\mathcal{F}}}$                             ▷ Sample new edges

8:      **for** $i = 1,\ldots,I$ **do**                             ▷ Loop from $i = 1$ up to $I$

9:          $\widetilde{O}^t_i := f_O\left(\widetilde{O}^{t-1}_i, \left\{\widetilde{F}^t_j\right\}_{j|(j,i)\in I^t}\right)$                ▷ Update the nodes

10:     **end for**

11:     $\widetilde{G}_{\mathcal{O}^1:\mathcal{O}^t} := \left(\widetilde{\mathcal{O}}^t\dot{\cup}\widetilde{G}^{\mathcal{V}}_{\mathcal{O}^1:\mathcal{F}^t}, \left\{E\left(\widetilde{F}^t_j, \widetilde{O}^t_i\right)\right\}_{(j,i)\in I^t}\dot{\cup}\widetilde{G}^{\mathcal{E}}_{\mathcal{O}^1:\mathcal{F}^t}\right)$    ▷ Update the graph

12:     $J^t \sim \rho^t_{\widetilde{\mathcal{O}}}$                             ▷ Sample new edges

13:     **for** $j = 1,\ldots,J$ **do**                             ▷ Loop from $j = 1$ up to $J$

14:         $\widetilde{F}^{t+1}_j := f_F\left(\widetilde{F}^t_j, \left\{\widetilde{O}^t_i\right\}_{i|(i,j)\in J^t}\right)$                ▷ Update the nodes

15:     **end for**

16: **end for**

17: **return** $\widetilde{G}_{\mathcal{O}^1:\mathcal{O}^T}$

---

