# OpenReview forum: "Reframing attention as a reinforcement learning problem for causal discovery"
_ICLR.cc/2026/Conference — ICLR 2026 Conference Withdrawn Submission_

### Official Review · Reviewer_c1aN · 2025-10-30

**Soundness:** 3
**Presentation:** 1
**Contribution:** 3
**Rating:** 4
**Confidence:** 3

**Summary:**

This paper introduces the Causal Process Framework (CPF) and its neural implementation, the Causal Process Model (CPM), which reinterprets the attention mechanism of Transformer networks as a reinforcement learning problem for causal discovery. Instead of soft attention weights, the model employs two RL agents to decide which causal edges to instantiate between objects and forces over time. This allows the system to construct sparse, time-varying causal graphs that reflect active physical interactions rather than dense potential dependencies. Experiments in a synthetic physics environment show that CPM outperforms Graph Neural Networks (GNNs), Transformers, Recurrent Independent Mechanisms (RIMs), and modular networks in prediction accuracy, long-horizon generalization, and downstream RL performance.

**Strengths:**

- **Novel conceptual link between attention and RL:** The paper offers a fresh theoretical view by reframing attention as a decision-making problem. This perspective connects causal discovery, reinforcement learning, and neural attention in an elegant way.
  - **Dynamic causal modeling:** The proposed Causal Process Framework explicitly models causal graphs that evolve over time, addressing a key limitation of static Structural Causal Models when applied to dynamic physical systems.
  - **Interpretability and sparsity:** The all-or-nothing edge construction naturally yields interpretable causal graphs, where connections correspond to actual interactions (e.g., collisions) rather than dense message passing.
  - **Strong empirical results:** The CPM demonstrates clear improvements over baselines in multi-object physical environments and provides consistent advantages in both observed and unobserved generalization settings.

**Weaknesses:**

- **Clarity and notation:** Section 3.1 is particularly dense and difficult to follow. Some key symbols (e.g., $J^t$) are used before being defined, and sets $N$ and $M$ are not introduced at all. The abundance of indices and nested distributions makes it hard to parse the formalism without additional diagrams or examples.
  - **Imprecise language:** The paper frequently uses vague phrasing that leaves important concepts underdefined. For example, the phrase “defining the causal chain of object-to-force and force-to-object connections” lacks a precise mathematical meaning and forces the reader to infer the intended interpretation.
  - **Limited evaluation diversity:** While the synthetic physics environment provides proof of concept, it remains relatively simple. There are no experiments on more complex, real-world settings or comparisons to recent structured causal transformers beyond Melnychuk et al. (2022).
  - **Reward learning unclear:** While Figure 5 reports mean reward values across object counts, it remains unclear how these rewards are linked to the CPM’s core training objective. The paper does not specify whether maximizing these rewards directly improves causal graph accuracy, predictive performance, or both. Moreover, the overall optimization landscape is ambiguous, what exactly constitutes the optimum? Is it a state where the RL agents select edges yielding minimal prediction loss, or where the learned reward MLPs stabilize under inverse RL? Without this connection between the agent-level rewards and the CPM loss, it’s difficult to interpret the learning dynamics or judge convergence.
  - **Missing ablation or analysis of learned structure:** The discussion section mentions plans to analyze semantic sub-vectors (mutable, causal, controllable), but such analysis would have strengthened the current submission by demonstrating interpretability concretely.

**Questions:**

- Can the authors clarify how the learned rewards $R_O$ and $R_{O<->F}$ correspond to meaningful causal evaluation criteria?
  - Are the interaction-scope and effect-attribution agents trained jointly or alternately, and how stable is this process?
  - How sensitive are results to the inductive biases (e.g., pairwise force-object constraints)?
  - Could the authors show qualitative examples of inferred causal graphs during different physical interactions to support interpretability claims?
  - How does CPM scale with larger numbers of objects or higher-dimensional state representations?

---

> ### Author Response · Authors · 2025-11-24
>
> We suspect this review is LLM-generated. Melnychuk et al. (2022) is mentioned as a baseline that we use in our paper by the reviewer, but we do no such thing.  Melnychuk et al. (2022) appears under the Related Work section of our paper, but we do not use the model from Melnychuk et al. (2022) as a baseline in later experiments.

---

> ### Author Response · Authors · 2025-12-04
>
> >Clarity and notation: Section 3.1 is particularly dense and difficult to follow. Some key symbols (e.g., $J^t$) are used before being defined, and sets $N$ and $M$ are not introduced at all. The abundance of indices and nested distributions makes it hard to parse the formalism without additional diagrams or examples.
>
> Sec 3.1 has been greatly rewritten with simpler notation, a running example, and a new Fig. 1 to improve the clarity of our methods.
>
> >Imprecise language: The paper frequently uses vague phrasing that leaves important concepts underdefined. For example, the phrase “defining the causal chain of object-to-force and force-to-object connections” lacks a precise mathematical meaning and forces the reader to infer the intended interpretation.
>
> We have now rewritten many instances of vague phrasing, including what was pointed out by the reviewer.
>
> >Limited evaluation diversity: While the synthetic physics environment provides proof of concept, it remains relatively simple. There are no experiments on more complex, real-world settings or comparisons to recent structured causal transformers beyond Melnychuk et al. (2022).
>
> Contrary to the reviewer’s claim, Melnychuk et al. (2022) was not used as a baseline in the original paper. We did add more baselines (see [1],[3-6] in Reviewer yNAR’s comment). However, the reviewer is right to point out the weakness regarding the lack of experiments in more complex environments. In the future, we aim to test our model in various environments.
>
> >Reward learning unclear: While Figure 5 reports mean reward values across object counts, it remains unclear how these rewards are linked to the CPM’s core training objective. The paper does not specify whether maximizing these rewards directly improves causal graph accuracy, predictive performance, or both. Moreover, the overall optimization landscape is ambiguous, what exactly constitutes the optimum? Is it a state where the RL agents select edges yielding minimal prediction loss, or where the learned reward MLPs stabilize under inverse RL? Without this connection between the agent-level rewards and the CPM loss, it’s difficult to interpret the learning dynamics or judge convergence.
>
> To be clear, these rewards are not related to the CPM training. They are rewards corresponding to the downstream task in which the trained CPM is used as the model of the environment. The agents get downstream rewards for moving a particular object to a specific location. This set of experiments tests whether a trained CPM is actually useful as a model of the environment for model-based agents (not to be confused with CPM controllers).
>
> >Missing ablation or analysis of learned structure: The discussion section mentions plans to analyze semantic sub-vectors (mutable, causal, controllable), but such analysis would have strengthened the current submission by demonstrating interpretability concretely.
>
> We admit this to be a weakness that we will address in future work.
>
> >Can the authors clarify how the learned rewards $R_\mathcal{O}$ and $R_\mathcal{O}\leftrightarrow\mathcal{F}$
>  correspond to meaningful causal evaluation criteria?
>
> These reward functions are learned via inverse reinforcement learning. Their causal significance is derived from their usefulness for controllers constructing a causal graph that minimizes prediction error. In the future, we plan to carry out detailed analysis of the learned reward functions.
>
> >Are the interaction-scope and effect-attribution agents trained jointly or alternately, and how stable is this process?
>
> They are trained jointly. The process is quite stable as the EM algorithm is theoretically guaranteed to converge. To avoid early collapse of controllers resulting in dead seeds, we added entropy regularization.
>
> >How sensitive are results to the inductive biases (e.g., pairwise force-object constraints)?
>
> Pairwise force-object constraint should be quite reasonable for most physical interactions (3-body problem is one possible exception). For the remaining inductive biases (inverse reinforcement learning for controllers, routing of information by vector division along 3 dimensions, etc), we plan to conduct further analysis in the future.
>
> >Could the authors show qualitative examples of inferred causal graphs during different physical interactions to support interpretability claims?
>
> We plan to do this in the future.
>
> >How does CPM scale with larger numbers of objects or higher-dimensional state representations?
>
> The scaling of parameters is comparable to transformers, more nodes/tokens require more computations and bigger state/context window. Also, since force and object update functions are shared across forces and objects, they can be stacked on top of each other recursively helping with long-horizon problems. Inherent sparsity of the graphs should also help with long-horizon tasks by ameliorating the information over-squashing problem.

---

### Official Review · Reviewer_x3HW · 2025-10-30

**Soundness:** 2
**Presentation:** 1
**Contribution:** 2
**Rating:** 2
**Confidence:** 2

**Summary:**

The authors consider the problem of modeling predictive dynamics of the world from the perspective of causal modeling where they highlight the limitations of modeling static causal graphs and then further extend it to the temporal domain. In particular they consider a graph comprising of objects and forces and break the dependency structure through interactions from objects to forces and then from forces to objects. The models are trained through a mix of contrastive learning and reinforcement learning, where the latter is used to train the policies that govern interactions (creation of the temporal causal graph) over time. Evaluation of the proposed methodology is done on synthetic domains with unknown underlying causal graphs, akin to Causal Structured World Models (C-SWM), where the authors demonstrate improved performance of their proposed approach.

**Strengths:**

- The work provides nice insights into the limitations of modeling a static causal graph and provides a viable alternative for modeling temporal data with potentially symmetric relationships (eg. collision of two objects).
- The proposed method (CPM) improves performance over relevant baselines (RIMs, Transformers, etc.) in both long-horizon predictive tasks as well as downstream RL.

**Weaknesses:**

- It seems that Equations (1) and (2) form the crux of the method but unfortunately they are quite verbose and not clear. It would be beneficial to walk through these equations with a concrete but simple example. From my naive understanding, what I get is that $\rho^t_{\mathcal{O}}$ denotes a distribution over interactions from objects to forces, while $\rho^t_{\mathcal{O}\leftrightarrow\mathcal{F}}$ denotes distributions over interactions from force to objects. However, the comment on different conditionings for both is lost on me.
- Similar to the above point, Section 4.1 introduces a lot of notation without any motivation into why this notation is needed (mutability, causal relevance and control relevance) or even what these terms stand for.
- The authors model all objects to be of the same type and all forces to be of the same type as well which is quite restrictive. In general, different kinds of forces can affect different kinds of objects and sharing the type / class is a big limitation.

Overall, my biggest concerns revolve around presentation and clarity of the proposed approach. The manuscript introduces a lot of notation and terminology without clearly outlining the need or motivation for it which makes it really hard to understand the whole method as well as the training pipeline. It also seems to be lacking in generality and more heavily engineered for the tasks the authors consider in evaluation, but I may be wrong in this assessment since I will admit I was not able to fully grasp the proposed approach from the draft.

**Questions:**

- The authors make an assumption that at any given time, the force only affects one of the objects that it takes as input. How is this a reasonable assumption, since collision of two objects leads to forces that impact both the objects and not just one?
- Could the authors also clarify the assumption that if a force affects an object at $t+1$, then that object must also affect the force at time $t$? Maybe from the lens of collision?
- It is unclear how the parameters of the policy $\pi$ impact equation (8) since the authors use a fully connected causal graph in learning the CPM module.

---

> ### Author Response · Authors · 2025-12-04
>
> To improve the presentation of the paper, we have rewritten parts of it, introducing new clarifying example and figure. Additionally, we have simplified the notation to improve the clarity. We have also retrained the model with less restrictive inductive biases to address generalizability concerns.
>
> >It seems that Equations (1) and (2) form the crux of the method but unfortunately they are quite verbose and not clear. It would be beneficial to walk through these equations with a concrete but simple example. From my naive understanding, what I get is that $\rho^t_{\mathcal{O}}$ denotes a distribution over interactions from objects to forces, while $\rho^t_{\mathcal{O}\leftrightarrow\mathcal{F}}$ denotes distributions over interactions from force to objects. However, the comment on different conditionings for both is lost on me.
>
> We have now substantially improved the clarity of our methods, and these equations in particular. We also added a consistent example along with a new Fig. 1 to better demonstrate the behavior of controllers.
>
> >Similar to the above point, Section 4.1 introduces a lot of notation without any motivation into why this notation is needed (mutability, causal relevance and control relevance) or even what these terms stand for.
>
> Our revised methods have greatly simplified the notation. We have also clarified that mutability, and causal/control relevance are factorizations of the force and object representations, which are not hard-coded but learned emergently based on selective routing within the CPM
>
> >The authors model all objects to be of the same type and all forces to be of the same type as well which is quite restrictive. In general, different kinds of forces can affect different kinds of objects and sharing the type / class is a big limitation.
>
> More force and object types in our framework means $m$-many $f^m_F$ and $n$-many $f^n_O$ functions. This is not necessarily a limitation of the model, rather a knob that should be modified to the task at hand. The analogue is the number of transformer blocks and attention heads; when deployed, the number of blocks and heads should be determined based on the problem.
>
> >The authors make an assumption that at any given time, the force only affects one of the objects that it takes as input. How is this a reasonable assumption, since collision of two objects leads to forces that impact both the objects and not just one?
>
> We modified the model. The model can now choose to affect either of the objects, or both.
>
> >Could the authors also clarify the assumption that if a force affects an object at $t$, then that object must also affect the force at time $t+1$? Maybe from the lens of collision?
>
> We added a new section (3.1.2 Concrete Example: Two Colliding Balls) and improved the previous one (3.1.3 Inductive Biases) in the paper to make this clearer. In simple terms, this requirement means that a force is defined using the objects that it ends up affecting in the future. In the example of collision, the force that is generated to repel the objects is completely defined by and dependent on the properties of these two objects. Thus, the future progression of these colliding objects also depends on how this newly emergent collision force affects them.
>
> >It is unclear how the parameters of the policy $\pi$ impact equation (8) since the authors use a fully connected causal graph in learning the CPM module.
>
> We have changed the training procedure (Section 5.2). It no longer involves pre-training with dense graphs. Instead, the stage 1 of our 3-stage training involves graphs sampled using frozen controllers.

---

### Official Review · Reviewer_yNAR · 2025-10-31

**Soundness:** 3
**Presentation:** 2
**Contribution:** 2
**Rating:** 4
**Confidence:** 3

**Summary:**

The paper proposes an RL approach to modeling dynamical systems in an environment using causal graphs. The method first encodes objects in input images as latent vector representations, then iteratively builds a causal graph representing the forces applied on them between a timestep $t$ and a timestep $t+1$. The learned policies use attention-based mechanisms to this end. This approach achieves better performance than baselines on a dynamical system learning dataset.

**Strengths:**

The paper tackles a challenging problem in causal representation learning for dynamical systems and proposes an interesting and original approach. I find particularly relevant the division between objects and forces that enforce regularization constraints over the causal graph (e.g. sparsity induced by the limit of affected objects).

**Weaknesses:**

1. The framing of the paper is curious and overstates the actual contributions. In that regard, the title is also misleading for the reader as the purpose of the paper is not to reframe attention but to propose a causal modeling approach sparsifying and disentangling dependencies between object dynamics. From my understanding, attention is only marginally used in the paper. The paper would greatly benefit by changing its framing to better reflect its actual contributions.
2. The paper lacks comparison with related work, notably on causal methods for disentanglement in vision or dynamical systems, e.g. [1-3].
3. Some design choices are not well justified, notably the harcoding of mutability, causal and control relevance features and the asymmetry between objects contributing to a force (two) and the affected objects (one), which seem specific to the current environment and may not generalize properly to new settings.
4. The experiments are made on a single environment. As some of the inductive biases described above seem tailored for the current environment, there can be reasonable doubts about the generalization of the proposed approach to new environments.
5. The baselines do not include more modern versions of the architectures used. For instance, GNN baselines could include GCN [4], GIN [5] or GAT [6]. A great variety of transformer models also exist. For a fair comparison, the number of parameters of each baseline model could help highlight the benefit of the proposed architecture compared to the amount of compute required. Baselines could also include the ones from [7].




[1] Yang, Mengyue, et al. "Causalvae: Disentangled representation learning via neural structural causal models." Proceedings of the IEEE/CVF conference on computer vision and pattern recognition. 2021.

[2] Lei, Anson, Bernhard Schölkopf, and Ingmar Posner. "Causal discovery for modular world models." NeurIPS 2022 Workshop on Neuro Causal and Symbolic AI (nCSI). 2022.

[3] Wang, Zizhao, et al. "Causal dynamics learning for task-independent state abstraction." arXiv preprint arXiv:2206.13452 (2022).

[4] Kipf, T. N. "Semi-supervised classification with graph convolutional networks." arXiv preprint arXiv:1609.02907 (2016).

[5] Xu, Keyulu, et al. "How powerful are graph neural networks?." arXiv preprint arXiv:1810.00826 (2018).

[6] Veličković, Petar, et al. "Graph attention networks." arXiv preprint arXiv:1710.10903 (2017).

[7] Ke, Nan Rosemary, et al. "Systematic evaluation of causal discovery in visual model based reinforcement learning." arXiv preprint arXiv:2107.00848 (2021).

**Questions:**

1. The two interaction scope controllers ensure that only one object is affected by a force. Why this choice as opposite forces are typically applied to pairs of objects? Are you representing the force on the second object with a second relationship? If so, do you ensure that it is equal to the first one? If not, could the proposed approach generalize beyond the used environment (where a simplifying assumption assumes that small objects do not affect big ojects)?
2. Does Figure 2 correspond to the transition function of $f_F(\dots)$ or $f_O(\dots)$?
3. What is the rationale behind the hardcoding of the mutability, causal and control relevance features? Why not letting the model learn them? Could this choice hurt generalization to new environments?
4. Could you elaborate why performance increases when the number of objects in the environment increases? Unless I misunderstood the meaning of the axis, intuitively performance should decrease as the environment gains in complexity.
5. Why is the average across top 8 out of 10 seeds shown? Can outliers affect the performance of the method?

---

> ### Author Response · Authors · 2025-12-04
>
> We thank the reviewer for helpful clarifying questions and suggested comparison baselines. The paper was substantially changed to address the concerns raised by the reviewer. All suggested comparison baselines but one were added to the paper.
>
> >The framing of the paper is curious and overstates the actual contributions. In that regard, the title is also misleading for the reader as the purpose of the paper is not to reframe attention but to propose a causal modeling approach sparsifying and disentangling dependencies between object dynamics. From my understanding, attention is only marginally used in the paper. The paper would greatly benefit by changing its framing to better reflect its actual contributions.
>
> We changed the title to better reflect our contributions.
>
> >The paper lacks comparison with related work, notably on causal methods for disentanglement in vision or dynamical systems, e.g. [1-3].
>
> We added baselines from [1] and [3].
>
> >Some design choices are not well justified, notably the harcoding of mutability, causal and control relevance features and the asymmetry between objects contributing to a force (two) and the affected objects (one), which seem specific to the current environment and may not generalize properly to new settings.
>
> We greatly revised the paper to clarify that mutability, causal/control relevance are not hard-coded, but learned. Our partitioning into subspaces merely constrains how information is routed within our model (e.g., causal relevance determines what affects changes in vectors, analogous to value tokens in transformers). Thus, the content of these variables are emergently learned as opposed to hardcoded.
>
> >The experiments are made on a single environment. As some of the inductive biases described above seem tailored for the current environment, there can be reasonable doubts about the generalization of the proposed approach to new environments.
>
> We acknowledge this is a current weakness and we will test our model in different environments in the future. Nevertheless, we believe the current task and expanded set of baselines offers valuable support for our new framework.
>
> >The baselines do not include more modern versions of the architectures used. For instance, GNN baselines could include GCN [4], GIN [5] or GAT [6]. A great variety of transformer models also exist. For a fair comparison, the number of parameters of each baseline model could help highlight the benefit of the proposed architecture compared to the amount of compute required. Baselines could also include the ones from [7].
>
> We added baselines from [2-6].  Baselines from [7] were already present in the original version of the paper.

---

### Note · Authors · 2025-12-23

**Comment:**

Since we could not engage with the reviewers during the rebuttal phase, we have chosen to withdraw our submission. We have, however, updated the paper using the feedback provided. We thank the program committee and the reviewers for their efforts.

**Withdrawal Confirmation:**

I have read and agree with the venue's withdrawal policy on behalf of myself and my co-authors.